# External validation of a claims-based model to predict left ventricular ejection fraction class in patients with heart failure

Mufaddal Mahesri[1], Kristyn Chin[1], Abheenava Kumar[2], Aditya Barve[3], Rachel Studer[4], Raquel Lahoz[4], Rishi J. Desai[1]*

1 Division of Pharmacoepidemiology and Pharmacoeconomics, Department of Medicine, Brigham and Women's Hospital & Harvard Medical School, Boston, MA, United States of America, 2 Novartis Healthcare Pvt. Ltd, Hyderabad, India, 3 Novartis Ireland Pvt. Ltd, Dublin, Ireland, 4 Novartis Pharma AG, Basel, Switzerland

* rdesai@bwh.harvard.edu

## Abstract

**Data Availability Statement:** Data for these analyses were made available to the authors through third-party license from IBM Truven, a commercial data provider in the US. As such, the

### Background

Ejection fraction (EF) is an important prognostic factor in heart failure (HF), but administrative claims databases lack information on EF. We previously developed a model to predict EF class from Medicare claims. Here, we evaluated the performance of this model in an external validation sample of commercial insurance enrollees.

### Methods

Truven MarketScan claims linked to electronic medical records (EMR) data (IBM Explorys) containing EF measurements were used to identify a cohort of US patients with HF between 01-01-2012 and 10-31-2019. By applying the previously developed model, patients were classified into HF with reduced EF (HFrEF) or preserved EF (HFpEF). EF values recorded in EMR data were used to define gold-standard HFpEF (LVEF $\geq$45%) and HFrEF (LVEF<45%). Model performance was reported in terms of overall accuracy, positive predicted values (PPV), and sensitivity for HFrEF and HFpEF.

### Results

A total of 7,001 HF patients with an average age of 71 years were identified, 1,700 (24.3%) of whom had HFrEF. An overall accuracy of 0.81 (95% CI: 0.80–0.82) was seen in this external validation sample. For HFpEF, the model had sensitivity of 0.96 (95%CI, 0.95–0.97) and PPV of 0.81 (95% CI, 0.81–0.82); while for HFrEF, the sensitivity was 0.32 (95%CI, 0.30–0.34) and PPV was 0.73 (95%CI, 0.69–0.76). These results were consistent with what was previously published in US Medicare claims data.

authors cannot make these data publicly available due to data use agreement. Other researchers can access these data by purchasing a license through Truven. Inclusion criteria specified in the Methods section would allow other researchers to identify the same cohort of patients we used for these analyses. Interested individuals may see https://marketscan.truvenhealth.com/marketscanportal/ for more information on accessing Truven data.

**Funding:** This study was supported by a collaborative research grant from Novartis Inc. The study was conducted by the authors independent of the sponsor. The funder provided support in the form of salaries for authors AK, AB, RS and RL but did not have any additional role in the study design, decision to publish, or preparation of the manuscript. The specific roles of these authors are articulated in the 'Author contributions' section.

**Competing interests:** Dr. Desai has received unrestricted research grants from Merck and Bayer for unrelated projects. Dr. Studer and Ms. Lahoz are employees of Novartis Pharma AG. Mr. Kumar is an employee of Novartis Healthcare Pvt. Ltd., India and Dr. Barve is an employee of Novartis Ireland Pvt. Ltd., Ireland. There are no conflicts of interest to disclose for the other co-authors.

## Conclusions

The successful validation of the Medicare claims-based model provides evidence that this model may be used to identify patient subgroups with specific EF class in commercial claims databases as well.

## Introduction

Ejection fraction (EF) is an important prognostic factor in heart failure (HF). HF with reduced ejection fraction (HFrEF) is well characterized and there are a number of evidence-based therapies available [1]. In contrast, HF with preserved EF (HFpEF) is more heterogeneous, poorly characterized and there are no approved therapies that improve outcomes [1].

Insurance claims databases allow for longitudinal follow-up at the patient level and are very useful in evaluation of disease epidemiology and treatment outcomes in routine care [2]. However, a major limitation with claims databases in studying HF is the lack of available results from procedures such as echocardiograms, which are used to measure EF. Consequently, one cannot directly distinguish between HFrEF and HFpEF based on administrative claims. To address this limitation, we previously developed a model to predict EF class using Medicare claims and validated using electronic medical record (EMR) data from two large healthcare provider networks from the Boston metropolitan area [3]. The primary objective of the current study was to evaluate the performance of this prediction model in an external validation cohort.

## Methods

### Data source

Claims data derived from the Truven MarketScan database linked to EMRs from the IBM Explorys database were used. Truven MarketScan covers 235 million lives of US citizens consisting of two core claims databases; 1) MarketScan Commercial Claims and Encounters—which contains healthcare data commercially insured individuals, encompassing employees, their spouses, and their dependents from the United States, 2) Medicare Supplemental and Coordination of Benefits—which contains the healthcare experiences of Medicare-eligible retirees with employer-sponsored Medicare Supplemental plans. Both these data sources contain longitudinally traceable information for their enrollees' medical diagnoses recorded with International Classification of Disease, 9[th] and 10[th] Clinical Modification (ICD-9/ICD-10 CM) codes, medical procedures recorded as Current Procedure Terminology (CPT) or ICD-9 procedure codes, and medication dispensing recorded using National Drug Codes (NDC). The IBM Explorys data platform is a data network that comprises integrated information from 360 hospitals and approximately 31,700 providers, covering approximately 50 million patient lives. The Explorys data has been used for multiple prior observational studies [4–7] and contain data derived from ambulatory electronic medical records (EMRs), inpatient EMRs, laboratory, pharmacy, health plans, billing and accounting, data warehouses, patient portals, satisfaction surveys, and care management systems. The Marketscan and Explorys linked population represent approximately 10% of the total MarketScan population.

### Study design

Adult patients were included in the study if they had ≥1 diagnosis code for HF (ICD-9 or ICD-10) from the Truven MarketScan claims database after 6 months of continuous

enrollment in their health plans and ≥1 recorded EF result, within 6 months prior or 1 month after the HF diagnosis date, from the IBM Explorys EMR database. The study period was between January 1$^{st}$ of 2012 and October 31 of 2019 and the HF diagnosis date successfully paired with a qualifying EF measurement was defined as the cohort entry date. The study protocol was approved by the Brigham and Women's Hospital Institutional Review Board. The Institutional Review Board committee waived the requirement for informed consent. This is a retrospective cohort study using a HIPAA de-identified dataset and individuals cannot be identified from the data.

## Model validation

A patient level analytic data file with information on the predictor variables (S1 Table) was created for the whole cohort of eligible HF patients from the Marketscan-Explorys linked dataset. All predictors were measured in the 6 months prior to and 1-month post cohort entry. Using the regression coefficients for each individual predictor variables and the y-intercept of the model, (as reported in Desai et al. [3]), we estimated the probabilities of patient belonging to HFrEF or HFpEF and classified patients into one of these two classes using the recommended cut off. We used EF data from IBM Explorys to define gold standard classification into HFpEF (LVEF ≥45%) and HFrEF (LVEF<45%). In case of ≥1 EF results, values recorded on days closest to the cohort entry dates were used to define the gold standard. The predicted classification was compared against the gold standard to complete this validation exercise.

## Statistical analysis

Patient characteristics including demographics, HF-related variables (e.g. diagnosis code recorded for HF, HF-related hospitalizations), HF-related medications and various co-morbid conditions (e.g. hyperlipidemia, hypertension, cardiomyopathy) were described stratified by HFrEF or HFpEF for this validation cohort. We calculated overall accuracy (correct classification rate = number of accurate predictions/number of total predictions), positive predictive value (probability of being a true case, given algorithm prediction) and sensitivity (the probability of being identified as a case of specific HF class by the algorithm for a true case out of the overall population) along with 95% confidence intervals. Further, the performance of this model was also tested in the following pre-specified subgroups: males and females, age <65 and > = 65 years, index date prior to October 2015 (ICD9 period) and after October 2015 (ICD10 period). It should be noted that we allowed multiple entries in the cohort, therefore some patients may contribute to both the ICD9 and ICD10 period subgroups. We also described patient characteristics in categories of patients accurately and inaccurately classified by our model to characterize misclassified populations.

## Results

### Study cohort

We identified 157,203 patients with at least 1 HF diagnosis following 6 months eligibility of continuous medical and pharmacy benefits. Of these patients, we included 7,001 who were at least 18 years old at cohort entry date and who had at least one EF result available between 180 days before and 30 days after index date. Details of the cohort construction are provided in Fig 1.

Table 1 contains data on baseline characteristics by EF class identified via the gold standard criteria using EMR-recorded EF values. We identified 5,301 patients as HFpEF and 1,700 patients as HFrEF. The average age was similar across both the groups (HFpEF = 71 years vs

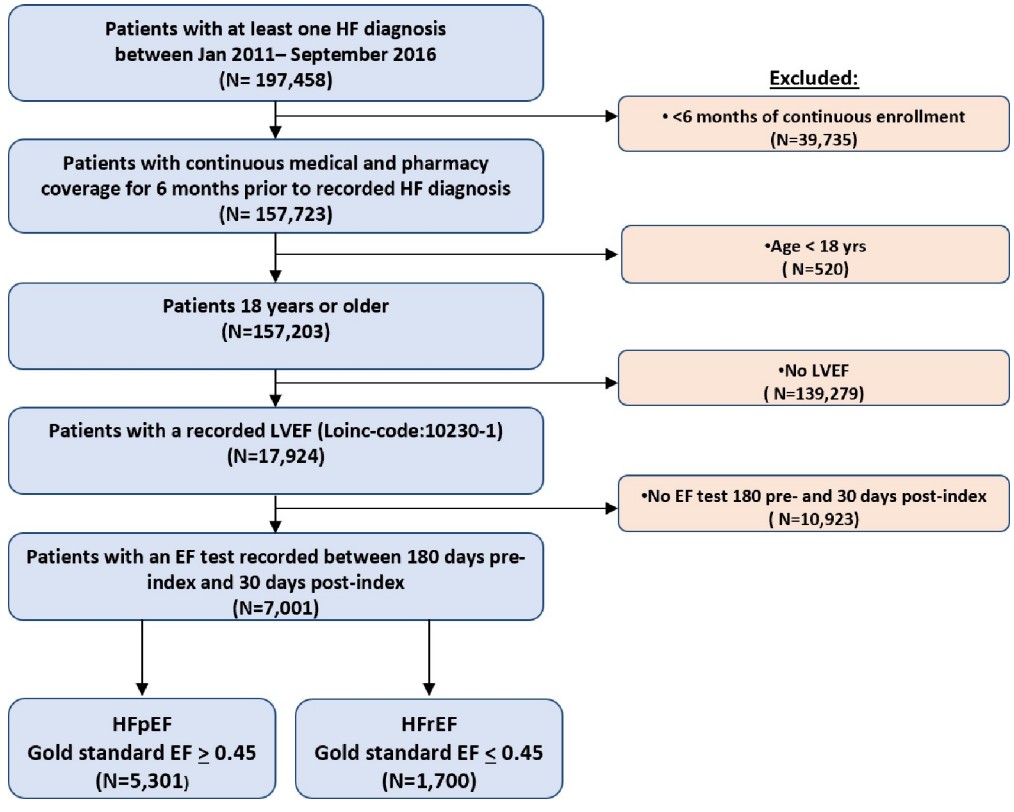

**Fig 1. Cohort consort diagram.**

HFrEF = 69 years) while males comprised 68% of HFrEF compared to 51% of HFpEF. The mean (SD) EF was 59%(7) in the HFpEF group while it was 32%(9) in the HFrEF group.

## Performance of the HF model

The model showed an overall accuracy of 0.81 (95% CI: 0.80–0.82). For HFpEF, the model had sensitivity of 0.96 (95% CI, 0.95–0.97) and PPV of 0.81 (95% CI, 0.81–0.82); while for HFrEF, the sensitivity was 0.32 (95% CI, 0.30–0.34) and PPV was 0.72 (95% CI, 0.69–0.76).

The overall accuracy was similar across the different subgroups; however, some variation was observed in sex subgroups. The overall accuracy was higher among female patients compared to male patients, due to a higher sensitivity and PPV in HFpEF. While, the male subgroup performed better for HFrEF. The model demonstrated very similar performance when using ICD-9 HF diagnoses compared to ICD-10 coded HF diagnoses. This was an important finding as the original model was developed using ICD-9 codes only and these finding support its use for data currently using both ICD-9 and ICD-10 diagnoses codes. Details of the performances of the primary model as well as the subgroup analyses are presented in Table 2. Patient characteristics in categories of patients accurately and inaccurately classified by our model are summarized for both HFrEF and HFpEF, in S2 Table.

## Discussion

As EF information is unavailable in administrative claims databases, it is important to develop claims-based models that can be used as a proxy to identify EF classes in patients with HF. In this external validation study, we assessed the accuracy of a claims-based model to predict EF

**Table 1. Baseline characteristics of HF patients stratified by ejection fraction class (HFrEF, < 0.45; or HFpEF, ≥ 0.45).**

| Variable | Gold standard HFrEF (N = 1,700) | Gold standard HFpEF (N = 5,301) |
|---|---|---|
| | N (%) | N (%) |
| Mean LVEF (in %), (SD) | 32 (9) | 59 (7) |
| **Demographics** | | |
| Male | 1152 (67.8) | 2687 (50.7) |
| Age in years, mean (SD) | 69.2 (14.0) | 70.6 (13.7) |
| **HF-related variables** | | |
| HF-specific ICD-9 and ICD-10 codes | | |
| Systolic HF | 657 (38.6) | 476 (9.0) |
| Diastolic HF | 83 (4.9) | 1360 (25.7) |
| Left HF | 94 (5.5) | 239 (4.5) |
| Unspecified HF | 790 (46.5) | 2930 (55.3) |
| HF Hospitalizations, mean (SD) | 0.2 (0.4) | 0.08 (0.3) |
| Implantable cardioverter-defibrillator | 245 (14.4) | 111 (2.1) |
| HF diagnosis identified in outpatient claims | 886 (52.1) | 3146 (59.3) |
| **HF-related medication use** | | |
| ACE inhibitors | 968 (56.9) | 2108 (39.8) |
| Mineralocorticoid receptor antagonists | 389 (22.9) | 467 (8.8) |
| Beta blockers | 998 (58.7) | 2587 (48.8) |
| Digoxin | 101 (5.9) | 118 (2.2) |
| Loop diuretics | 952 (56.0) | 2489 (46.9) |
| Nitrates | 285 (16.8) | 519 (9.8) |
| Thiazide diuretics | 629 (37.0) | 1581 (29.8) |
| **Comorbidities** | | |
| Atrial fibrillation or flutter | 723 (42.5) | 1956 (36.9) |
| Anemia | 583 (34.3) | 2121 (40.0) |
| Coronary artery bypass graft | 132 (7.8) | 292 (5.5) |
| Cardiomyopathy | 789 (46.4) | 572 (10.8) |
| Chronic obstructive pulmonary disease | 422 (24.8) | 1539 (29.0) |
| Depression | 209 (12.3) | 941 (17.7) |
| Hypertensive nephropathy | 241 (14.2) | 772 (14.6) |
| Hyperlipidemia | 1063 (62.5) | 3356 (63.3) |
| Hypertension | 1365 (80.3) | 4375 (82.5) |
| Hypotension | 293 (17.2) | 811 (15.3) |
| Myocardial infarction | 436 (25.6) | 608 (11.5) |
| Obesity | 324 (19.1) | 1277 (24.1) |
| Other dysrhythmias | 1002 (58.9) | 2469 (46.6) |
| Psychosis | 539 (31.7) | 1964 (37.0) |
| Rheumatic heart disease | 260 (15.3) | 994 (18.7) |
| Sleep apnea | 235 (13.8) | 950 (17.9) |
| Stable angina | 215 (12.6) | 540 (10.2) |
| Valve disorders | 278 (16.3) | 1148 (21.7) |

class developed in Medicare data, by applying it to commercial claims data to establish generalizability of this model outside of Medicare claims.

The performance with commercial claims was noted to be equivalent to the performance previously reported for the internal validation sample using Medicare claims [3]. In this study, we observed sensitivity of 0.96 and PPV of 0.81 in identifying HFpEF patients. This is very

**Table 2. Primary analysis and subgroup- specific performance.**

| Analysis | Overall Accuracy With 95% CIs | Reduced Ejection Fraction | | Preserved Ejection Fraction | |
|---|---|---|---|---|---|
| | | Positive Predicted Value With 95% CIs | Sensitivity With 95% CIs | Positive Predicted Value With 95% CIs | Sensitivity With 95% CIs |
| Primary analysis | 0.81 (0.80–0.82) | 0.72 (0.69–0.76) | 0.32 (0.30–0.34) | 0.81 (0.81–0.82) | 0.96 (0.95–0.97) |
| Subgroup 1: Age 65–75 y | 0.80 (0.78–0.82) | 0.73 (0.66–0.80) | 0.32 (0.28–0.37) | 0.81 (0.79–0.83) | 0.96 (0.95–0.97) |
| Subgroup 2: Age 75 y and older | 0.80 (0.79–0.82) | 0.73 (0.66–0.79) | 0.20 (0.17–0.23) | 0.81 (0.79–0.82) | 0.98 (0.97–0.98) |
| Subgroup 3: Males | 0.77 (0.75–0.78) | 0.73 (0.69–0.77) | 0.35 (0.32–0.38) | 0.77 (0.76–0.79) | 0.95 (0.94–0.95) |
| Subgroup 4: Females | 0.85 (0.84–0.86) | 0.70 (0.63–0.76) | 0.27 (0.23–0.30) | 0.86 (0.85–0.88) | 0.98 (0.97–0.98) |
| Subgroup 5: Entry HF diagnosis in inpatient claims | 0.80 (0.78–0.81) | 0.76 (0.72–0.80) | 0.37 (0.34–0.40) | 0.80 (0.78–0.82) | 0.96 (0.95–0.96) |
| Subgroup 6: Entry HF diagnosis in outpatient claims | 0.81 (0.80–0.82) | 0.68 (0.63–0.73) | 0.28 (0.25–0.31) | 0.83 (0.81–0.84) | 0.96 (0.96–0.97) |
| Subgroup 7: ICD-9 coded HF | 0.80 (0.78–0.81) | 0.72 (0.66–0.77) | 0.28 (0.25–0.32) | 0.80 (0.79–0.82) | 0.96 (0.96–0.97) |
| Subgroup 8: ICD-10 coded HF | 0.79 (0.78–0.80) | 0.72 (0.68–0.75) | 0.34 (0.31–0.36) | 0.80 (0.79–0.81) | 0.95 (0.95–0.96) |

similar to what was reported by Desai et al. in Medicare claims data (sensitivity of 0.97, PPV of 0.84). For HFrEF patients a substantially lower sensitivity (0.32) and a relatively lower PPV (0.72) was seen, which is also consistent with what was previously published (sensitivity of 0.29, PPV of 0.73).

We want to emphasize certain cautions that must be weighed carefully when using this model to identify EF classes in HF. First, the low sensitivity in identifying HFrEF would result in a considerable amount of sample being lost. Further, the group that is identified as HFrEF may systematically differ than the group that is misclassified by the model. On comparing the accurately classified HFrEF patients (547) with the misclassified HFrEF patients (1,153), we observed that EF was lower in accurately classified patients (average of 29% versus 33%, S2 Table). Compared to the misclassified HFrEF patients, the accurately classified HFrEF patients showed a higher prevalence of HF-related comorbidities, such as cardiomyopathy (85% versus 28%), myocardial infarction (34% versus 21%) and other dysrhythmias (67% versus 55%). Thus, patients identified as HFrEF by this model represents a sicker group. However, despite the low sensitivity, a recent study describing the epidemiology of HFrEF patients, identified in claims data based on this algorithm, showed characteristics and outcome trajectories for HFrEF patients that closely resembled other well-characterized population-based cohorts [8].

Determining EF class based on information routinely available in electronic data sources has received increasing recognition in recent years to address frequent unavailability of EF values. A retrospective cohort study among Minnesota residents evaluated a claims-based approach to identify HFpEF patients based on HF diagnosis codes in combination with laboratory orders for BNP/NT-proBNP and achieved PPV of 84% [9]. The higher PPV for HFpEF reported in our study is likely explained by addition of frequent comorbid conditions and demographics in addition to diagnoses codes. A second study by Uijl et al. based on data from the Swedish Heart Failure Registry used an approach similar to ours, where 22 predictors including laboratory results such as NT-proBNP, renal function; demographics such as age, sex; and comorbid conditions were used to classify patients into HFpEF and HFrEF [10]. The authors noted discrimination of 0.73 for this model in an external validation cohort. Results from our study along with the study by Uijl et al. suggest that model-based computable phenotyping of HF patients may provide a useful way to identify and study HF subtypes from electronic healthcare sources where EF values are not available. Our model may be more applicable in the context of US insurance claims databases, which often lack information such as laboratory test results.

Some limitations deserve mention. Although EMR data includes rich clinical information, high amount of missing data is to be expected. Consequently, generalizability may be limited if the patients with recorded EF values are not representative of the full HF population. Further, in the clinical setting, the diagnosis of HFpEF is typically a diagnosis of exclusion and may require confirmatory information about structural changes of the heart, beyond EF alone. Consequently, even though EF improves the accuracy of the diagnosis, there might be false positive HFpEF patients. Finally, we used a cutpoint of 45% to differentiate between 45% to differentiate between HFrEF and HFpEF, which meant we did not attempt to identify moderately reduced (mr) EF (40–49%). As noted by Desai et al. the EF cutpoint of 45% being implemented in our model is based on two major considerations. Firstly, a cutoff at either end of the 40% or 50% EF range could combine all patients with mrEF (40–49%) within a single group which could meaningfully change distribution of patient characteristics selectively in one of these two groups and make prediction difficult. Secondly, as EF changes over time are common in both pEF and rEF, a cutpoint at the mid-point of the mrEF range (45%) might capture more accurately those mrEF patients who either have recovered EF from initial rEF or reduced EF from initial pEF. Moreover, the 45% cutoff has also been used in multiple pivotal trials of HF patients (e.g TOPCAT trial for pEF patients and VICTORIA trial for rEF patients) [11, 12].

In conclusion, results from this study provide evidence regarding the generalizability of an approach using claims data to identify EF classes in HF patients outside of Medicare claims. This will aid future studies evaluating health outcomes, healthcare utilization as well as cost of care among HF patients in routine care when EF measurements are not available.

## Supporting information

**S1 Fig. Receiver operating characteristics (ROC) curve for the model.**
(TIF)

**S1 Table. Operational definitions for the variables included in the EF class prediction algorithm.**
(PDF)

**S2 Table. Baseline characteristics of HF patients correctly and incorrectly classified by algorithm compared to gold standard classification.**
(PDF)

## Author Contributions

**Conceptualization:** Mufaddal Mahesri, Rachel Studer, Raquel Lahoz, Rishi J. Desai.

**Data curation:** Abheenava Kumar, Aditya Barve.

**Formal analysis:** Abheenava Kumar, Aditya Barve.

**Funding acquisition:** Rishi J. Desai.

**Investigation:** Mufaddal Mahesri, Kristyn Chin, Rachel Studer, Raquel Lahoz, Rishi J. Desai.

**Methodology:** Mufaddal Mahesri, Kristyn Chin, Rachel Studer, Raquel Lahoz, Rishi J. Desai.

**Project administration:** Rishi J. Desai.

**Supervision:** Rishi J. Desai.

**Writing – original draft:** Mufaddal Mahesri, Kristyn Chin, Abheenava Kumar, Aditya Barve, Rachel Studer, Raquel Lahoz, Rishi J. Desai.

**Writing – review & editing:** Mufaddal Mahesri, Kristyn Chin, Abheenava Kumar, Aditya Barve, Rachel Studer, Raquel Lahoz, Rishi J. Desai.

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
