## [Decision Letter · Decision Letter 0]

19 Feb 2021

PONE-D-20-39176

External validation of a claims-based model to predict left ventricular ejection fraction class in patients with heart failure

PLOS ONE

Dear Dr. Desai,

Thank you for submitting your manuscript to PLOS ONE. After careful consideration, we feel that it has merit but does not fully meet PLOS ONE’s publication criteria as it currently stands. Therefore, we invite you to submit a revised version of the manuscript that addresses the points raised during the review process.

We look forward to receiving your revised manuscript.

Kind regards,

Gianluigi Savarese

Academic Editor

PLOS ONE

Journal Requirements:

"This study was supported by a collaborative research grant from Novartis Inc. The study was conducted by the authors independent of the sponsor. The sponsor was given the opportunity to make nonbinding comments on a draft of the manuscript, but the authors retained the right of publication and determined the final wording."

We note that one or more of the authors have an affiliation to the commercial funders of this research study : [Novartis Healthcare Pvt. Ltd, Novartis Ireland Pvt. Ltd, Novartis Pharma AG].

3.1. Please provide an amended Funding Statement declaring this commercial affiliation, as well as a statement regarding the Role of Funders in your study. If the funding organization did not play a role in the study design, data collection and analysis, decision to publish, or preparation of the manuscript and only provided financial support in the form of authors' salaries and/or research materials, please review your statements relating to the author contributions, and ensure you have specifically and accurately indicated the role(s) that these authors had in your study. You can update author roles in the Author Contributions section of the online submission form.

3.2. Please also provide an updated Competing Interests Statement declaring this commercial affiliation along with any other relevant declarations relating to employment, consultancy, patents, products in development, or marketed products, etc.  

Reviewers' comments:

Reviewer's Responses to Questions

**Comments to the Author**

1. Is the manuscript technically sound, and do the data support the conclusions?

Reviewer #1: Yes

Reviewer #2: Yes

Reviewer #3: Partly

2. Has the statistical analysis been performed appropriately and rigorously? 

Reviewer #1: Yes

Reviewer #2: Yes

Reviewer #3: No

3. Have the authors made all data underlying the findings in their manuscript fully available?

Reviewer #1: No

Reviewer #2: No

Reviewer #3: Yes

4. Is the manuscript presented in an intelligible fashion and written in standard English?

Reviewer #1: Yes

Reviewer #2: Yes

Reviewer #3: Yes

5. Review Comments to the Author

Reviewer #1: Mahesri and coworkers describe the external validation of a previously developed prediction model for LV ejection fraction in a claims-based dataset. The paper is topical and of interest to improve the 'researchability' of claims datasets. I have some points of concern.

1. Heart failure and left ventricular ejection fraction are intertwined for historical reasons. The cut-off of either below or above 45% is questioned, at least in European ESC/HFA guideline and/or position papers in recent years. I understand it is not feasible to predict other LVEF ranges since this is a validation of a previously developed model. However, it deserves attention in the discussion since many peers view LVEF range 40-50% as mildy reduced, with more resemblance of HFrEF <40%, at least in response to drugs.

2. In addition to first point, could there be many patients within LVEF 40-50% which resulted in decreased sensitivity and PPV? The model could discriminate more easily between the two extremes, as stated by the authors. What I am missing in the discussion are explanations for the poor model performance for HFrEF. The question of interest for this paper is not as to whether it is feasible to externally validate the model (i.e. present conclusion of this ms), but given the sens+PPV, what is the usabilty of this model in daily research practise? I struggle to be convinced that a sensitivity of 30% for HFrEF, in a highly selected population (i.e. 17 924 out of 157 203 patients for which LVEF was available) will yield sensible results if this model was to be used on claims datasets.

3. Consider giving just one decimal in table 1 for the percentages.

4. Please shortly mention the variables used by the model to predict the two phenotypes of HFrEF and HFpEF.

5. minor; introduction section: cardiac cahteterization is rarely used to estimate LV ejection fraction nowadays, at least in Western European countries.

Reviewer #2: In this interesting analysis the authors confirm the discriminatory ability of the model they have constructed to classify patients captured in administrative claims databases into HFrEF (herein defined as left ventricular ejection fraction <45%) and HFpEF (herein defined as left ventricular ejection fraction ≥45%). I have several objections regarding the value of the study. I am skeptical on whether a HF classification with 45% used as an EF cut-off is clinically meaningful. Furthermore, the low sensitivity of the model to identify HFrEF (32%) and the considerable misclassification seen with the model for both HFpEF (19%) and HFrEF (27%) increases the degree of uncertainty and bias which are already and issue in case of administrative database. I, therefore, doubt as to the potential utility of this model to support reliable clinical research.

Nonetheless, the study presents the results of original research, which has not been published elsewhere. The methods are described in detail and the analyses are performed to a high technical standard and described in sufficient detail. The conclusions are supported by the data and presented appropriately. The article is presented in an intelligible fashion and is written in standard English.

As the latter are the sole criteria based on which a manuscript submitted to this journal should be evaluated, I find that the paper fulfils all of them and merits publication.

Reviewer #3: The aim of this study was to externally validate a model which predicts reduced and preserved ejection fraction (EF) in heart failure in a sample of commercial insurance enrollees. This represents a different domain of patients which includes those <65 years old and beyond patients who were under Medicare coverage (in which the original model was developed).

The research is relevant to the prediction of EF status in electronic health or claims databases which lacks information on EF values.

It is good to see that overall accuracy, sensitivity, specificity, and PPV for HFrEF and HFpEF here remained similar to the original population when tested in a heart failure population with lower disease severity and fewer comorbid conditions such as previous myocardial infarction, anaemia and atrial fibrillation. While these metrics are commonly used to indicate performance of prediction models, their values are dependent on the health setting and prevalence of the condition of interest in the study population. For instance, higher disease prevalence will raise the PPV and decrease the NPV.

What would be necessary to assess the performance of this model compared to the original one is the model discrimination; the trade-off between sensitivity and specificity. The c-statistics provided in the model development paper was 0.86 (35-predictor binomial model), what is the c-statistics and how does the discrimination plot/ ROC curve look like in this external validation cohort?

Also, it would be important to know the calibration of the model, which refers to the extent of agreement between the predicted probabilities of the model versus the observed frequencies. A calibration plot would be helpful for readers to determine how close the model predictions are to the observed gold standard frequencies.

The following references may come handy:

-DOI: 10.1136/heartjnl-2011-301247

-doi: 10.1093/ckj/sfaa188

-http://dx.doi.org/10.1136/bmj.i3140

Please describe the predictors used in the model in methods, I counted 34 predictors in the Appendix but the original model uses 35? If predictors were not available in the new dataset, it would be helpful to explain in the Methods.

Please also discuss the study findings in relation to existing/previous work by others

In conclusion, the findings from this study is of interest to future work on phenotyping HF patients in large administrative databases

6. PLOS authors have the option to publish the peer review history of their article (what does this mean?). If published, this will include your full peer review and any attached files.

Reviewer #1: No

Reviewer #2: No

Reviewer #3: No

---

## [Author Response · Author response to Decision Letter 0]

5 May 2021

Response to Reviewers– Manuscript ID: PONE-D-20-39176 – PLOS ONE

External validation of a claims-based model to predict left ventricular ejection fraction class in patients with heart failure

We greatly appreciate the opportunity to revise this manuscript. Please see below for our point-by-point responses and the revised manuscript with tracked changes.

Journal Requirements:

Response: Thank you. The manuscript was revised to meet the PLOS ONE style requirements. The files have been renamed as appropriate. 

2. Please provide additional details regarding participant consent. 

Response: Thank you. We have included the following statement in the Methods section:

“The Institutional Review Board committee waived the requirement for informed consent. This is a retrospective cohort study using a HIPAA de-identified dataset and individuals cannot be identified from the data.” 

3.1. Please provide an amended Funding Statement declaring this commercial affiliation, as well as a statement regarding the Role of Funders in your study. 

Response: Thank you for your comment. Based on the instructions, we have updated our Funding Statement to declare affiliation to the commercial funders for some of the authors and updated the authors’ roles. We have added the following:

“This study was supported by a collaborative research grant from Novartis Inc. The study was conducted by the authors independent of the sponsor. The funder provided support in the form of salaries for authors AK, AB, RS and RL but did not have any additional role in the study design, decision to publish, or preparation of the manuscript. The specific roles of these authors are articulated in the ‘author contributions’ section.” 

3.2. Please also provide an updated Competing Interests Statement declaring this commercial affiliation along with any other relevant declarations relating to employment, consultancy, patents, products in development, or marketed products, etc. 

Response: We have updated our Competing Interest Statement to confirm that this commercial affiliation does not alter our adherence to PLOS ONE policies on sharing data and materials. We have added the following text in the Competing Interest Statement:

“Dr. Desai has received research grants from Merck and Bayer to the Brigham and Women’s Hospital for projects outside the submitted work. Dr. Studer and Ms. Lahoz are employees of Novartis Pharma AG. Mr. Kumar is an employee of Novartis Healthcare Pvt. Ltd., India and Dr. Barve is an employee of Novartis Ireland Pvt. Ltd., Ireland. There are no conflicts of interest to disclose for the other co-authors. This does not alter our adherence to PLOS ONE policies on sharing data and materials.” 

4. We note that you have indicated that data from this study are available upon request.

Response: Data for these analyses were made available to the authors through third-party license from Truven, a commercial data provider in the US. As such, the authors cannot make these data publicly available due to data use agreement. Other researchers can access these data by purchasing a license through Truven. Inclusion criteria specified in the Methods section would allow other researchers to identify the same cohort of patients we used for these analyses. Please see https://marketscan.truvenhealth.com/marketscanportal/ for more information on accessing Truven data.

Response to comments from Reviewers:

Reviewer #1: Mahesri and coworkers describe the external validation of a previously developed prediction model for LV ejection fraction in a claims-based dataset. The paper is topical and of interest to improve the 'researchability' of claims datasets. I have some points of concern.

1. Heart failure and left ventricular ejection fraction are intertwined for historical reasons. The cut-off of either below or above 45% is questioned, at least in European ESC/HFA guideline and/or position papers in recent years. I understand it is not feasible to predict other LVEF ranges since this is a validation of a previously developed model. However, it deserves attention in the discussion since many peers view LVEF range 40-50% as mildy reduced, with more resemblance of HFrEF <40%, at least in response to drugs

Response: Thank you, and this is an important question. First, we would re-iterate that given somewhat distinct patterns of characteristics and outcomes in patients with moderately reduced (mr) EF (40-49%) observed in previous research [Nadruz et al. Circ Heart Fail. 2016 Apr;9(4):e002826], it would be ideal to have an algorithm that can identify these patients separately from rEF and pEF patients. However, as noted in the original publication that describes the development of the claims-based model [Desai et al. Circ Cardiovasc Qual Outcomes. 2018;11(12):e004700.], the algorithm did not succeed in identifying these patients with reliable accuracy. As a result, we had to determine an EF cutpoint for the binary model. We chose the cutpoint of 45% based on the following two reasons: 1) we reasoned that separating out patients at either end of the mrEF range (40 or 50%) could combine all mrEF patients with a single group (with pEF for 40% cutoff or with rEF for 50% cutoff) group, which could meaningfully change distribution of patient characteristics selectively in one of these two groups and make prediction difficult, and 2) as noted in previous research [Dunlay et al. Circ Heart Fail. 2012;5:720-726], EF changes over time are common in both pEF and rEF. We thought that placing the cutpoint at the mid-point of the mrEF range (45%) might capture those mrEF patients who either have recovered EF from initial rEF or reduced EF from initial pEF with some accuracy in their EF class. Additionally, as noted in the study by Desai et al., the 45% cutoff has also been used in multiple pivotal trials of HF patients (e.g TOPCAT trial for pEF patients and VICTORIA trial for rEF patients). Therefore, this choice is not completely arbitrary. We have added the following text to summarize this rationale in the discussion (page 11):

“As noted by Desai et al. the EF cutpoint of 45% being implemented in our model is based on two major considerations. Firstly, a cutoff at either end of the 40% or 50% EF range could combine all patients with moderately reduced (mr) EF (40-49%) within a single group which could meaningfully change distribution of patient characteristics selectively in one of these two groups and make prediction difficult. Secondly, as EF changes over time are common in both pEF and rEF, a cutpoint at the mid-point of the mrEF range (45%) might capture more accurately those mrEF patients who either have recovered EF from initial rEF or reduced EF from initial pEF. Moreover, the 45% cutoff has also been used in multiple pivotal trials of HF patients (e.g TOPCAT trial for pEF patients and VICTORIA trial for rEF patients).”

2. In addition to first point, could there be many patients within LVEF 40-50% which resulted in decreased sensitivity and PPV? The model could discriminate more easily between the two extremes, as stated by the authors. What I am missing in the discussion are explanations for the poor model performance for HFrEF. The question of interest for this paper is not as to whether it is feasible to externally validate the model (i.e. present conclusion of this ms), but given the sens+PPV, what is the usabilty of this model in daily research practise? I struggle to be convinced that a sensitivity of 30% for HFrEF, in a highly selected population (i.e. 17 924 out of 157 203 patients for which LVEF was available) will yield sensible results if this model was to be used on claims datasets.

Response: Thank you. We recognize the limitation and are transparent in describing the low sensitivity for rEF patients in the discussion section. Having said this, in claims data, a large proportion of HF patients are coded as having “unspecified HF”. This inherent limitation of the data contributes to the low model performance for HFrEF patients in claims data. In a recent study, we have characterized the epidemiology of claims-identified HFrEF patients based on this algorithm and have noted that despite the low sensitivity, characteristics and outcome trajectories of patients identified as HFrEF based on this model closely resemble well-characterized population-based cohorts. We have added the following to the discussion section:

“However, despite the low sensitivity, a recent study describing the epidemiology of HFrEF patients, identified in claims data based on this algorithm, showed characteristics and outcome trajectories for HFrEF patients that closely resembled other well-characterized population-based cohorts [8].

Reference:

[8] Desai RJ, Mahesri M, Chin K, Levin R, Lahoz R, Studer R, Vaduganathan M, Patorno E. Epidemiologic Characterization of Heart Failure with Reduced or Preserved Ejection Fraction Populations Identified Using Medicare Claims. Am J Med. 2020 Oct 27:S0002-9343(20)30924-4. doi: 10.1016/j.amjmed.2020.09.038. Epub ahead of print. PMID: 33127370]. 

3. Consider giving just one decimal in table 1 for the percentages

Response: Thank you for your comment. We have made this change in the manuscript.

4. Please shortly mention the variables used by the model to predict the two phenotypes of HFrEF and HFpEF

Response: Thank you. The definitions of the predictor variables are included in the Supporting information, S-1 Table. 

5. minor; introduction section: cardiac cahteterization is rarely used to estimate LV ejection fraction nowadays, at least in Western European countries

Response: Thank you for your comment. We have removed “cardiac catheterization” from the second paragraph in the introduction.

Reviewer #2: In this interesting analysis the authors confirm the discriminatory ability of the model they have constructed to classify patients captured in administrative claims databases into HFrEF (herein defined as left ventricular ejection fraction <45%) and HFpEF (herein defined as left ventricular ejection fraction ≥45%). I have several objections regarding the value of the study. I am skeptical on whether a HF classification with 45% used as an EF cut-off is clinically meaningful. Furthermore, the low sensitivity of the model to identify HFrEF (32%) and the considerable misclassification seen with the model for both HFpEF (19%) and HFrEF (27%) increases the degree of uncertainty and bias which are already and issue in case of administrative database. I, therefore, doubt as to the potential utility of this model to support reliable clinical research.

Nonetheless, the study presents the results of original research, which has not been published elsewhere. The methods are described in detail and the analyses are performed to a high technical standard and described in sufficient detail. The conclusions are supported by the data and presented appropriately. The article is presented in an intelligible fashion and is written in standard English.

As the latter are the sole criteria based on which a manuscript submitted to this journal should be evaluated, I find that the paper fulfils all of them and merits publication.

Response: Thank you for your comment. Please refer to our response to Reviewer 1, comment 1 and comment 2 for a detailed discussion of the points raised.

Reviewer #3: The aim of this study was to externally validate a model which predicts reduced and preserved ejection fraction (EF) in heart failure in a sample of commercial insurance enrollees. This represents a different domain of patients which includes those <65 years old and beyond patients who were under Medicare coverage (in which the original model was developed).

The research is relevant to the prediction of EF status in electronic health or claims databases which lacks information on EF values.

It is good to see that overall accuracy, sensitivity, specificity, and PPV for HFrEF and HFpEF here remained similar to the original population when tested in a heart failure population with lower disease severity and fewer comorbid conditions such as previous myocardial infarction, anaemia and atrial fibrillation. While these metrics are commonly used to indicate performance of prediction models, their values are dependent on the health setting and prevalence of the condition of interest in the study population. For instance, higher disease prevalence will raise the PPV and decrease the NPV.

1. What would be necessary to assess the performance of this model compared to the original one is the model discrimination; the trade-off between sensitivity and specificity. The c-statistics provided in the model development paper was 0.86 (35-predictor binomial model), what is the c-statistics and how does the discrimination plot/ ROC curve look like in this external validation cohort?

Response: We appreciate the comment. The c-statistic for our model was 0.83 (95% CI: 0.81-0.84) which is comparable to the c-statistics reported in the model development paper by Desai et al. We have included the c-statistic along with the following ROC curve graph in the appendix of the manuscript:

Appendix S-3 Figure. Receiver operating characteristics (ROC) curve for the model.

2. Also, it would be important to know the calibration of the model, which refers to the extent of agreement between the predicted probabilities of the model versus the observed frequencies. A calibration plot would be helpful for readers to determine how close the model predictions are to the observed gold standard frequencies.

Response: Thank you for the comment. We have included the calibration plot below for the reviewer. However, as our model is describing a classification exercise and not a prediction exercise, we do not feel the calibration plot adds any additional value here since the predicted probability values are only used to inform the cutoff for classifying rEF and pEF. Therefore, we have opted to not include it in the manuscript. 

Calibration plot for the model

3. Please describe the predictors used in the model in methods, I counted 34 predictors in the Appendix but the original model uses 35? If predictors were not available in the new dataset, it would be helpful to explain in the Methods.

Response: Thank you. All 35 predictor variables in the original model are included in the model presented here. The y-intercept value was included as the 35th predictor variable in both the original and this model. We have clarified this in the text as well.

4. Please also discuss the study findings in relation to existing/previous work by others

In conclusion, the findings from this study is of interest to future work on phenotyping HF patients in large administrative databases.

Response: Thank you for your comment. The following discussion was added:

“Determining EF class based on information routinely available in electronic data sources has received increasing recognition in recent years to address frequent unavailability of EF values. A retrospective cohort study among Minnesota residents evaluated a claims-based approach to identify HFpEF patients based on HF diagnosis codes in combination with laboratory orders for BNP/NT-proBNP and achieved PPV of 84% [9]. The higher PPV for HFpEF reported in our study is likely explained by addition of frequent comorbid conditions and demographics in addition to diagnoses codes. A second study by Uijil et al. based on data from the Swedish Heart Failure Registry used an approach similar to ours, where 22 predictors including laboratory results such as NT-proBNP, renal function; demographics such as age, sex; and comorbid conditions were used to classify patients into HFpEF and HFrEF [10]. The authors noted discrimination of 0.73 for this model in an external validation cohort. Results from our study along with the study by Uijil et al. suggest that model-based computable phenotyping of HF patients may provide a useful way to identify and study HF subtypes from electronic healthcare sources where EF values are not available. Our model may be more applicable in the context of US insurance claims databases, which often lack information such as laboratory test results.”

References:

[9] Cohen JB, Schrauben SJ, Zhao L, Basso MD, Cvijic ME, Li Z, Yarde M, Wang Z, Bhattacharya PT, Chirinos DA, Prenner S, Zamani P, Seiffert DA, Car BD, Gordon DA, Margulies K, Cappola T, Chirinos JA. Clinical Phenogroups in Heart Failure With Preserved Ejection Fraction: Detailed Phenotypes, Prognosis, and Response to Spironolactone. JACC Heart Fail. 2020 Mar;8(3):172-184. doi: 10.1016/j.jchf.2019.09.009. Epub 2020 Jan 8.

[10] Uijl A, Lund LH, Vaartjes I, Brugts JJ, Linssen GC, Asselbergs FW, Hoes AW, Dahlström U, Koudstaal S, Savarese G. A registry-based algorithm to predict ejection fraction in patients with heart failure. ESC Heart Fail. 2020 Oct;7(5):2388-2397. doi: 10.1002/ehf2.12779. Epub 2020 Jun 17.

---

## [Decision Letter · Decision Letter 1]

25 May 2021

External validation of a claims-based model to predict left ventricular ejection fraction class in patients with heart failure

PONE-D-20-39176R1

Dear Dr. Desai,

We’re pleased to inform you that your manuscript has been judged scientifically suitable for publication and will be formally accepted for publication once it meets all outstanding technical requirements.

Kind regards,

Gianluigi Savarese

Academic Editor

PLOS ONE

Additional Editor Comments (optional):

Reviewers' comments:

Reviewer's Responses to Questions

**Comments to the Author**

1. If the authors have adequately addressed your comments raised in a previous round of review and you feel that this manuscript is now acceptable for publication, you may indicate that here to bypass the “Comments to the Author” section, enter your conflict of interest statement in the “Confidential to Editor” section, and submit your "Accept" recommendation.

Reviewer #1: All comments have been addressed

Reviewer #2: All comments have been addressed

2. Is the manuscript technically sound, and do the data support the conclusions?

Reviewer #1: Yes

Reviewer #2: Yes

3. Has the statistical analysis been performed appropriately and rigorously? 

Reviewer #1: Yes

Reviewer #2: Yes

4. Have the authors made all data underlying the findings in their manuscript fully available?

Reviewer #1: Yes

Reviewer #2: Yes

5. Is the manuscript presented in an intelligible fashion and written in standard English?

Reviewer #1: Yes

Reviewer #2: Yes

6. Review Comments to the Author

Reviewer #1: The revised manuscript has been considerably improved and is well balanced. I have no further comments.

Reviewer #2: All previous comments have been addressed in detail by the authors. I have no further changes to request.

7. PLOS authors have the option to publish the peer review history of their article (what does this mean?). If published, this will include your full peer review and any attached files.

Reviewer #1: No

Reviewer #2: No

---

## [Editor Report · Acceptance letter]

28 May 2021

PONE-D-20-39176R1 

External validation of a claims-based model to predict left ventricular ejection fraction class in patients with heart failure 

Dear Dr. Desai:

I'm pleased to inform you that your manuscript has been deemed suitable for publication in PLOS ONE. Congratulations! Your manuscript is now with our production department. 

Kind regards, 

on behalf of

Dr. Gianluigi Savarese 

Academic Editor

PLOS ONE